# Yth m^6^A RNA-Binding Protein 1 Regulates Osteogenesis of MC3T3-E1 Cells under Hypoxia via Translational Control of Thrombospondin-1

**DOI:** 10.3390/ijms24021741

**Published:** 2023-01-16

**Authors:** Diwen Shi, Xiaohan Liu, Xinyun Li, Tian Li, Jie Liu, Lin Wu

**Affiliations:** 1Department of Prosthodontics, School of Stomatology, China Medical University, Shenyang 110000, China; 2Centre of Science Experiment, China Medical University, Shenyang 110000, China

**Keywords:** hypoxia, osteoblast differentiation, YTHDF1, THBS1, mRNA stability, peri-implantitis

## Abstract

Peri-implantitis is a major factor affecting implant prognosis, and the specific anatomy of the peri-implant area makes it more vulnerable to the local hypoxic environment caused by inflammation. N6-methyladenosine (m^6^A) plays a vital role in a multitude of biological processes, and its main “reader” Yth m^6^A RNA-binding protein 1 (YTHDF1) is suggested to affect osteogenic differentiation. However, the mechanism underlying the effect of YTHDF1 on osteogenic differentiation under hypoxic conditions remains unclear. To address this question, we examined the expression of YTHDF1 under hypoxia and observed that hypoxia suppressed osteogenic differentiation but promoted the expression of YTHDF1. Then we knocked down YTHDF1 and found decreased levels of osteogenic-related markers, alkaline phosphatase (ALP) activity, and alizarin red staining (ARS) under normoxia or hypoxia treatment. Bioinformatics analysis identified Thrombospondin-1 (THBS1) might be a downstream factor of YTHDF1. The results revealed that YTHDF1 enhanced the stability of THBS1 mRNA, and immunofluorescence assays found co-localization with YTHDF1 and THBS1 under hypoxia. Loss of function studies showed knocking down YTHDF1 or THBS1 exacerbated the osteogenic inhibition caused by hypoxia. All data imply that hypoxia suppresses osteogenic differentiation and promotes the expression of YTHDF1, which translationally regulates THBS1 in an m^6^A-dependent manner, potentially counteracting hypoxia-induced osteogenic inhibition through the YTHDF1/THBS1 pathway. The results of this study reveal for the first time the molecular mechanism of the regulation of osteogenic differentiation by YTHDF1 under hypoxia and suggest that YTHDF1, together with its downstream factor THBS1, may be critical targets to counteract osteogenic inhibition under hypoxic conditions, providing promising therapeutic strategy for the hypoxia-induced bone loss in peri-implantitis.

## 1. Introduction

Peri-implantitis is a pathological state of peri-implant tissue caused by bacterial plaque, which is the main cause of dental implant failure [1]. Studies have reported that within 5–10 years of dental implant placement, 20% of patients suffered implant loss related to infection [2]. Inflammation and hypoxia are closely correlated, in that inflammatory injuries usually cause hypoxic changes [3]. Hypoxia is defined as a systemic or local depletion of oxygen and triggers certain events within the body in response to oxygen deficiency [4]. Tissue changes related to hypoxia are known to occur in gingivitis and periodontitis [5,6], yet little is known about the changes in peri-implant tissues. Both Márcia’s and Ozkan Karata’s studies have shown the presence of hypoxic environments surrounding implants [7,8]. Due to the absence of two vital components of periodontal tissue: junctional epithelium and periodontal membrane, as well as cytoarchitecture and vascular structures, peri-implant tissues exhibit a higher risk of hypoxia [9]. Compared with natural teeth, the host response and immune cell migration are more restricted in peri-implant tissues [10]. In addition, the loading pressure generated by implant placement affects the blood circulation in periodontal tissues [7], and the absence of a periodontal membrane causes poor blood vessel formation, leading to further oxygen deprivation of tissues [11]. Therefore, peri-implant tissues may be more vulnerable to inflammation and hypoxia than natural teeth, and further exploration of the regulating mechanism under hypoxia would be helpful to clinical peri-implantitis treatment.

N6-methyladenosine (m^6^A) RNA methylation is an emergent molecular mechanism that can regulate gene expression at the post-transcriptional level [12], and is also the most common epigenetic modification in eukaryotic mRNAs and modulates a wide range of biological processes, such as cellular stress responses and viral infections [13]. The m^6^A methylation modification process is dynamic and reversible, with methylation transferases (Writers)[14], and demethylases (Erasers) demethylating the m^6^A -modified bases [15]. Specific RNA-binding proteins (RBP), also known as “readers”, interact with gene transcripts and influence mRNA maturation, transport, stability, and degradation [16]. They can recognize and bind to m^6^A -modified transcripts at the mRNA level, and mediate a variety of processes of mRNA metabolism, such as mRNA transcription, splicing, and stability [17]. Studies have shown that in the hypoxic environment generated by circumstances, such as hepatocellular carcinoma, breast cancer, and islet transplantation, Yth m^6^A RNA-binding protein 1 (YTHDF1) appeared higher expression relative to normal tissues [18,19,20]. Therefore, we speculate that YTHDF1 expression may be upregulated in the hypoxic environment resulting from dental implant placement.

In recent years, numerous researchers have pointed out the importance of m^6^A modification in regulating the process of osteogenesis [21,22,23]. A recent study revealed that YTHDF1 facilitated osteogenic differentiation in vitro and in vivo. He et al. found that methyltransferase-like 3 (METTL3) could regulate chondrocyte differentiation through dentin matrix protein 1 mRNA by YTHDF1-mediated methylation [24]. Nevertheless, little is known about the function of YTHDF1 on osteogenesis in hypoxic environments, and it is also unknown whether YTHDF1 affects the cellular hypoxic responses and its regulatory pathways in osteoblasts. Therefore, our study aimed to investigate the underlying regulatory mechanism of YTHDF1 on osteogenesis under hypoxia. According to the findings of our results, we hypothesized that YTHDF1 regulates Thrombospondin-1 (THBS1) post-transcriptionally through m^6^A methylation to promote osteogenic differentiation of MC3T3-E1 cells, providing an important potential for the treatment of peri-implantitis.

## 2. Results

### 2.1. Hypoxia Promotes YTHDF1 Expression and Inhibits Osteoblasts Differentiation

To investigate the role of YTHDF1 in peri-implantitis, we retrieved microarray data related to peri-implantitis using the gene expression omnibus (GEO) database and finally obtained GSE33774 microarray data, which showed a significant increase in YTHDF1 expression in peri-implantitis tissue (Figure 1A). MC3T3-E1 cells were treated at three different oxygen concentrations (21%, 5%, 1%) to simulate the hypoxia environment during the peri-implantitis pathogenesis. Compared to the normoxia group (21% oxygen concentrations), hypoxia increased the expression of hypoxia-inducible factor-1α (HIF-1α), which is the most sensitive indicator of the degree of cellular hypoxia. Both 5% and 1% oxygen concentrations upregulated Hif-1α expression, and to the greatest extent at 1% oxygen concentration (Figure 1B). The expression level of YTHDF1 was found to be significantly upregulated in the 1% O_2_ group and peaked at 24 h in contrast to the 0 h group. The mRNA and protein expression of HIF-1α was significantly upregulated in a time-dependent manner at 1% O_2_, while YTHDF1 expression peaked at 24 h (Figure 1C,D). Taken together, we exposed the cells in hypoxic conditions at 1% O_2_ concentration for 24 h in further experiments. Hypoxic culture resulted in a decrease in the mRNA expression of osteogenic differentiation-related marker genes, including *Alpl*, *Runx2*, and *Col1a1* in MC3T3-E1 cells after 24 h cultured under hypoxic conditions (Figure 1E), and the protein level of key transcription factors alkaline phosphatase (ALP), runt-related transcription factor 2 (RUNX2), type 1 collagen (COL1) were also confirmed the osteogenic repression by hypoxia (Figure 1F). In addition, ALP activity assay and ARS staining for extracellular matrix mineralization (EMM) were used to analyze the osteogenic differentiation of MC3T3-E1 cells under normoxic and hypoxic culture during the process of differentiation. As shown by both ALP activity and alizarin red staining (ARS), hypoxic treatment reduced the osteogenesis of MC3T3-E1 cells (Figure 1G). These results implied that hypoxia decreased osteogenic differentiation while promoting the expression of YTHDF1 in MC3T3-E1 cells.

### 2.2. Hypoxia Promotes YTHDF1 Expression and Inhibits Osteoblasts Differentiation

To determine the role of the m^6^A methylation reader YTHDF1 as a regulator of bone formation under hypoxic conditions, we performed loss-of-function experiments in MC3T3-E1 cells. YTHDF1 expression was suppressed with short interfering RNA (siRNA). When YTHDF1 was silenced, YTHDF1 protein and mRNA levels were remarkably knocked down compared to the si-NC (Figure 2A,B). We evaluated the knockdown efficiency of the three siRNAs, and the si-Ythdf1-3 with the best efficiency was utilized in the following experiments. Both in normoxic and hypoxic conditions, YTHDF1 knockdown resulted in the downregulation of mRNA and protein expression of YTHDF1 in MC3T3-E1 cells, whereas under hypoxic conditions, YTHDF1 expression appeared to increase compared to normoxia (Figure 2C,D). Under normoxic conditions, lessened protein expression levels of osteogenic differentiation-related markers ALP, RUNX2, and COL1 in YTHDF1 knockdown MC3T3-E1 cells were also verified by Western blot (Figure 2E), and silenced YTHDF1 led to a significant decrease in mRNA expression of *Alpl*, *Runx2*, and *Col1a1* in MC3T3-E1 cells after 24 h culture (Figure 2F). The mRNA expression of osteoblast-specific genes declined significantly compared to the normoxia groups, with the most evident decrease in the YTHDF1 knockdown group under hypoxia. The staining intensities of ALP and ARS were also weakened in the MC3T3-E1 cells of the YTHDF1 knockdown groups compared to the control groups, with YTHDF1 silencing under hypoxia showing the lowest staining intensity (Figure 2G,H). The data from the loss-of-function experiments indicated that YTHDF1 facilitated the osteogenic differentiation of MC3T3-E1 cells under both normoxia and hypoxia.

### 2.3. Bioinformatics Analysis Identified THBS1 as a Downstream Factor of YTHDF1

To investigate the potential mechanism of YTHDF1 in the osteogenesis downregulation induced by hypoxia, we performed bioinformatics analysis. Interactions between YTHDF1 and altered m^6^A peaks during osteogenesis were analyzed using the sequencing results of hBMSCs [25]. Considering the dynamic regulation of m^6^A acts in osteogenic differentiation, m^6^A peaks with abundance changed during osteogenesis were taken to intersect with those in genes regulated by YTHDF1. According to the acquired results and data obtained from the GO database, the results of GO biological process (BP) analysis of YTHDF1 with osteogenic acting target genes are shown in Figure 3A, which revealed that genes intersection were significantly enriched in biological processes of response to hypoxia, negative regulation of transcription from RNA polymerase II promoter and chromatin organization. The results showed enrichment to five genes associated with the hypoxic response: *MECP2*, *PRKAA1*, *THBS1*, *TM9SF4*, *HK2*, and *THBS1* was the gene with m^6^A modification increased the most to be recognized by YTHDF1.

We then attempted to explore whether THBS1 expression is modulated by YTHDF1 in an m^6^A-dependent manner. First, we used the mammalian m^6^A locus predictor SRAMP to search for m^6^A sites in THBS1. SRAMP analysis revealed that the m^6^A modification motifs of THBS1 mRNA have a wide distribution (Figure 3B), with three very high confidence m^6^A sites at locus 1222, 4283, and 4392 on CDS. PRIdictor and PRISeq analyses indicate that YTHDF1 has strong potential to be combined with THBS1 mRNA (Figure 3C), and the binding area intended to be coherent with the distribution of THBS1 mRNA m^6^A modification motifs. Based on the above considerations, we hypothesized that THBS1 is an m^6^A-modified gene bound by YTHDF1 and functions as a regulator in the osteogenesis of MC3T3-E1 cells, thus requiring further investigation.

### 2.4. THBS1 mRNA Regulated by YTHDF1 in a M^6^A-Dependent Way

We further attempted to confirm whether THBS1 was regulated by YTHDF1. The mRNA and protein expression levels of THBS1 in control and YTHDF1-silenced MC3T3-E1 cells were assessed 48 h after transfection, and the results showed that silencing YTHDF1 downregulated the mRNA and protein expression of THBS1 under normoxia and hypoxia (Figure 3D). Silencing of YTHDF1 in the presence of the transcriptional inhibitor actinomycin D (Act D) significantly reduced the stability of THBS1 mRNA both in normoxic and hypoxic conditions (Figure 3E), while hypoxia fractionally restored the stability of THBS1 mRNA. These results indicate that the regulation of THBS1 by YTHDF1 occurs post-transcriptionally.

Furthermore, we assumed that YTHDF1 could co-localize with THBS1 in MC3T3-E1 cells under hypoxia. To obtain the precise localization of YTHDF1 and THBS1, we also conducted immunofluorescence (IF) assays and found that under normoxia conditions, the THBS1 protein was located in the nucleus while YTHDF1 protein was located in the cytoplasm. After 24 h of being cultured under hypoxia conditions, the YTHDF1 expression increased compared to the control cells and was found co-localized with THBS1 in the cytoplasm (Figure 3F). These results suggest that YTHDF1 may be related to the osteogenesis of MC3T3-E1 cells under hypoxia through post-transcriptional regulation of the mRNA of THBS1.

### 2.5. THBS1 Promotes Osteogenic Differentiation of MC3T3-E1 Cells under Hypoxia

We have demonstrated the promotional effect of YTHDF1 on the osteogenic differentiation of MC3T3-E1 cells under hypoxia, and the above experiments predicted THBS1 as a downstream target of YTHDF1. Subsequently, we applied a loss-of-function approach to determine the effect of THBS1 on osteogenesis under normoxic and hypoxic conditions. After transfection of si-Thbs1 for 48 h, the knockdown efficiency was verified by qRT-PCR and Western blot analysis under both normoxia and hypoxia (Figure 4A). Hypoxia boosted THBS1 expression, and silencing of THBS1 knocked down the expression level of THBS1 under both normoxia and hypoxia. During osteogenic differentiation of MC3T3-E1 cells, silencing of THBS1 significantly suppressed the mRNA expression and protein levels of osteogenic-related ALP, RUNX2, and COL1 under hypoxia, and the knockdown of THBS1 under hypoxia showed the most significant inhibition of osteogenesis (Figure 4C,D). Results of ALP and ARS staining experiments also indicated that knocking down of THBS1 declined the mineralization of MC3T3-E1 cells under normoxia and hypoxia, with the lowest staining intensity shown in cells transfected with si-Thbs1 under hypoxia (Figure 4E,F).

### 2.6. THBS1 as a Target Gene of YTHDF1 and a Mediator of the Osteogenic Differentiation under Hypoxia

Overexpression of YTHDF1 was achieved by transfection of Ythdf1-oe plasmid into MC3T3-E1 cells. The transfection efficiency was validated by qRT-PCR and Western blot analysis 48 h after Thbs1-oe transfection (Figure 5A,B). Our previous analysis predicted that YTHDF1-dependent m^6^A methylation of THBS1 mRNAs correlates with osteogenic differentiation of MC3T3-E1 cells. The results verified that overexpressed YTHDF1 promoted YTHDF1 and THBS1 mRNA expression under both normoxia and hypoxia, and the extent of enhancement was significantly higher under hypoxia (Figure 5C). The co-localization of YTHDF1 and THBS1 after overexpression of YTHDF1 also corroborated the results of qRT-PCR (Appendix A). In contrast to knockdown experiments, overexpression of YTHDF1 optimized the expression of protein related to osteogenic differentiation (Figure 5D). Next, we went on to provide further insight into the molecular mechanisms underlying our findings. After performing co-transfection of si-Thbs1 with Ythdf1-oe on MC3T3-E1 cells, we examined the protein expression level of osteogenesis-related genes under normoxia and hypoxia. Consistent with our hypothesis, the Western blot revealed that under both normoxic and hypoxic conditions, silencing THBS1 reverted the upregulation of osteogenesis-related protein expression of ALP, RUNX2, and COL1 caused by YTHDF1 overexpression (Figure 5E). ALP staining results also indicated that after co-transfection of si-Thbs1 with overexpressed YTHDF1, the osteogenesis-enabling effect of YTHDF1 overexpression was attenuated (Figure 5F). These results suggest that MC3T3-E1 cells may promote osteogenesis through the YTHDF1/THBS1 pathway, either by upregulating the expression of YTHDF1 or THBS1, respectively, to resist hypoxia-induced osteogenic inhibition.

## 3. Discussion

Peri-implantitis is a major risk factor affecting the prognosis of implants [26]. Local inflammatory pathological changes in the periodontium induced by bacteria can trigger hypoxic responses in the periodontal supporting tissues by promoting the release of inflammatory factors [27], and the absence of the periodontal membrane reduces the local blood supply, aggravating the tissue hypoxia and causing bone resorption [11]. Hypoxia can cause a series of biological responses in bone tissue metabolism, mainly manifested as bone tissue resorption [28]. Therefore, further investigation of the mechanisms affecting osteogenesis in the hypoxic environment is essential to control the development of peri-implantitis. HIF-1α is one of the most sensitive factors in response to hypoxia and is highly activated at low oxygen concentrations [29]. It was utilized as an indicator of the degree of cellular hypoxia. In this study, we identified 1% O_2_ culture for 24 h treatment was used to simulate severe hypoxia in peri-implantitis. Here, we observed that hypoxia decreased the expression of osteogenesis-related genes of MC3T3-E1 cells, and ALP activity and osteoblast mineralization were also decreased. In addition, the m^6^A reader YTHDF1 was increased under hypoxic stress, but facilitated osteogenic differentiation of MC3T3-E1 cells, indicating that it may promote bone regeneration through post-transcriptional regulation of specific methylation-modified mRNAs.

N6-methyladenosine methylation is one of the most abundant modifications at the RNA level and performs a key role in gene regulation, affecting mRNA decay, splicing, translocation, localization, and translation [16]. Accumulating evidence indicates that m^6^A binding protein plays a vital role in m^6^A modification [30]. As the most widely recognized methylase reader, YTHDF1 has been investigated in many scientific research fields. Recent studies have shown a strong correlation between hypoxia and YTHDF1, and clinical evidence proves that YTHDF1 is associated with cancer progression and can maintain hypoxia tolerance by promoting the translation of certain proteins [31,32,33]. It has been shown that YTHDF1 expression upregulated in a HIF-1α-dependent manner by promoting transcription in human hepatocellular carcinoma [18]. Moreover, YTHDF1 has shown to be capable of promoting osteogenic differentiation of hBMSC both in vivo and in vitro, and its downstream ZNF839 may function as a co-activator of Runx2 [25]. Our findings revealed the role of YTHDF1 in regulating the osteogenesis of MC3T3-E1 cells and suggested that YTHDF1 may be a potential therapeutic target for peri-implantitis treatment. The results revealed that YTHDF1 promoted osteogenesis after either normoxic or hypoxic treatment, and the loss-of-function experiments under hypoxia confirmed that silencing YTHDF1 further exacerbated hypoxia-induced osteogenic inhibition in comparison to the normoxic control, suggesting a positive effect of YTHDF1 on osteoblast differentiation in a hypoxic environment.

In this study, we identified THBS1 as a downstream target of YTHDF1 by bioinformatics analysis. THBS1 is known as a glycoprotein and also acts as an endogenous angiogenesis inhibitor [34]. THBS1 level dramatically elevates under hypoxia and is implicated in several cardiovascular disorders [35]. Previous studies have shown that THBS1 is associated with endothelial dysfunction by inducing the onset of inflammatory responses, and is involved in platelet aggregation and adhesion to endothelial cells [36]. Yet its effects related to osteogenic differentiation remain to be investigated. It has been reported that THBS1 can promote the proliferation of MC3T3-E1 cells [37], and THBS1 in canine endothelial colony-forming cell (ECFC) exosomes can mediate angiogenesis and osteogenic differentiation in distraction osteogenesis via the PI3K/AKT/ERK pathway [38]. Although the role of THBS1 in regulating osteogenic differentiation has not been detailed in the literature, its expression elevated under hypoxia, and showed potential m^6^A methylation binding sites with YTHDF1, prompting us to speculate that it may act as a downstream factor of YTHDF1 and functions in osteogenesis of MC3T3-E1 cells. Moreover, our reversion experiments also validated this osteogenic enhancement, where the knockdown of THBS1 under hypoxia exacerbated the inhibitory effect of hypoxia on osteogenic differentiation. The results of our ALP and ARS assays and qRT-PCR analysis indicated that THBS1 can potentiate the osteogenesis of MC3T3-E1 cells under normoxia and hypoxia. In this study, the protein THBS1 was post-transcriptionally controlled by YTHDF1 in an m^6^A-dependent manner. The immunofluorescence results indicated that the THBS1 expression was elevated under hypoxia, and the hypoxia-induced THBS1 expression transferred from the nucleus to the cytoplasm, which suggests hypoxia may induce methylated THBS1 mRNA to exit the nucleus and be post-transcriptionally modified by YTHDF1. Under such a theme, we propose that YTHDF1 potentiates the osteogenesis of MC3T3-E1 cells by stabilizing the mRNA of THBS1, thereby promoting its translation and increasing the expression level of THBS1 to prevent osteogenesis inhibition caused by hypoxia, as shown schematically in Figure 6.

The limitation of our study is that the downstream targets of YTHDF1 may not be limited to THBS1, and further studies that explore other downstream targets of YTHDF1 are required for a better understanding of the role of YTHDF1 in the pathogenesis of peri-implantitis. Moreover, the mechanism of THBS1 in promoting osteogenesis under hypoxia requires further exploration. In addition, in vivo studies to validate our findings should be performed in further experiments.

## 4. Materials and Methods

### 4.1. Cell Culture and Hypoxia Treatment

The murine osteoblastic MC3T3-E1 cell line obtained from the Central Laboratory, School of Stomatology, China Medical University (Shenyang, China) was cultured with alpha-minimum essential medium (α-MEM; Biological Industries, Kibbutz Beit Haemek, Israel), supplemented with 10% thermo-inactivated fetal bovine serum (FBS; TianHangBiotech, Zhejiang, China), and 1% penicillin–streptomycin (Gibco; Carlsbad, CA, USA). Cells culture was undertaken in a humidified 37 °C incubator containing a 5% CO_2_ atmosphere until they reached confluence. To investigate the appropriate hypoxia condition, MC3T3-E1 cells were randomly divided into 3 groups: 1% oxygen content (severe hypoxia group), 5% oxygen content (slight hypoxia group), and 21% oxygen content (control group). Moreover, 2 groups of hypoxic treatment were cultivated in a 3-gas (N_2_/O_2_/CO_2_) incubator (Thermo Scientific, Wilmington, DE, USA).

### 4.2. Transient Transfection

Cells were cultured in six-well plates (4 × 10^5^ cells/well), and were transfected until the cell density could reach 70% confluency after 18–24 h. Then, 20 nM of small interfering RNA (siRNA) were transiently transfected into MC3T3-E1 cells using lipofectamine 2000 following the manufacturer’s instruction. siRNA-targeted sequences were listed in Table 1.

### 4.3. Total RNA Extraction and Quantitative Real-Time PCR

Total RNA was extracted with RNAiso Plus Reagent (Takara Bio, Tokyo, Japan) and dissolved in RNase-free water. Reverse transcribed using the Reverse Transcription System (Takara Bio, Tokyo, Japan). PCR amplification was performed using SYBR Premix Ex Taq™ II (Takara Bio, Tokyo, Japan) and the CFX-96 quantitative PCR system (Bio-Rad Laboratories, Inc, Hercules, CA, USA) after the extraction procedure. The β-actin gene was used as the reference gene and three independent sets of experiments were carried out independently to ensure accuracy. Relative mRNA levels of each marker were calculated by the 2-ΔΔ Ct method after normalizing to β-actin. The primers used in this step are listed in Table 2.

### 4.4. mRNA Stability

MC3T3-E1 cells were cultured into six-well plates (4 × 10^5^ cells/well) and transfected with 5 μg/mL actinomycin D (APExBIO Technology, Houston, TX, USA) at 0, 6, and 12 h, respectively to inhibit mRNA transcription [39]. Total RNAs were extracted for qRT-PCR analysis.

### 4.5. Western Blotting

After being washed with pre-chilled phosphate-buffered saline (PBS), cells were lysed in RIPA buffer (Beyotime, Shanghai, China) supplemented with protease inhibitors and allosteric nucleases. Protein samples were quantified by BCA kit (Beyotime, Shanghai, China). Approximately 20 μg of proteins were separated using 10% SDS-PAGE gel electrophoresis and transferred to polyvinylidene difluoride membranes (PVDF membranes). The membranes were blocked with 5% non-fat milk and incubated at 4 °C in anti- YTHDF1 (1:1000; 66745-1-Ig, Proteintech, Wuhan, China), THBS1 (1:1000; 18304-1-AP, Proteintech, Wuhan, China), GAPDH (1:4000; AF7021, Affinity, USA), HIF-1α(1:500; WL01607, Wanlei, Shenyang, China), ALP (1:1000; DF6225, Affinity, USA), RUNX2 (1:1000; PB0171, Boster, Wuhan, China), COL1(1:1000; AF7001, Affinity, USA) overnight at 4 °C, and sequentially with secondary antibody goat anti-mouse (1:10,000; SA00001-1, Proteintech, Wuhan, China), and goat anti-rabbit (1:10,000; SA00001-2, Proteintech, Wuhan, China) for 1 h at room temperature. Finally, the membranes were washed with 1% TBST. Visualization of blots was performed with Bio-Rad ChemiDoc XRS+ Imaging System (USA) and ECL kit (Biosharp, Hefei, China) according to the manufacturer’s instructions. For each blot, the relative density was quantified with Image J software (NIH, Bethesda, MD, USA).

### 4.6. Immunofluorescence Assays

The cells were seeded in a 30-mm confocal dish. After being cultured for 24 h, the dishes were fixed with 4% paraformaldehyde for 20 min and permeabilized in 0.5% Triton for 15 min. Afterward, the samples were blocked with 5% bovine serum albumin for 1 h and probed by specific primary antibodies for YTHDF1 (1:100; Proteintech, Wuhan, China), THBS1(1:100; Proteintech, Wuhan, China) at 4 °C overnight after treatment. Later they were detected with fluorescent secondary antibodies (1:100; Thermo Scientific, Wilmington, DE, USA) in a dark room, followed by 4′,6-diamidino-2-phenylindole (DAPI) (Beyotime, Shanghai, China) for 2 min. The samples were visualized and photographed with an inverted confocal microscope (Olympus, Tokyo, Japan).

### 4.7. ALP Assay

For ALP staining, after 7 days of being cultured in normoxia or hypoxia, cells were washed thrice with deionized water after discarding the medium and fixed with 4% polyoxymethylene. Then, the cells were incubated at RT in the dark with an ALP assay kit (Beyotime, Shanghai, China). After 30 min of incubation, the cells were washed three times with deionized water and observed under standard light microscopy.

### 4.8. Alizarin Red Staining

Alizarin Red staining was used to assess calcium-rich deposits in MC3T3-E1 cells, after 21 days of osteogenic induction, cells were fixed with 4% polyoxymethylene for 15 min and stained with 0.1% alizarin red (Solarbio; Beijing, China) at RT for 30 min. The cells were washed thrice with deionized water after staining and observed with light microscopy, and the alizarin red-positive area in each well was calculated to evaluate cell mineralization.

### 4.9. Bioinformatics Analysis

We used the m^6^A modification site predictor SRAMP (http://www.cuilab.cn/sramp/, accessed on 11 February 2022) to predict the distribution of the m^6^A modification motif of the target mRNA. PRIdictor (http://bclab.inha.ac.kr/pridictor/, accessed on 24 April 2022)) was applied to predict the binding sites of proteins to the target mRNA, and RMbase v2.0 (http://rna.sysu.edu.cn/rmbase/, accessed on 7 March 2022)), RM2Target (http://rm2target.canceromics.org/, accessed on 7 March 2022))was employed to identify the major m^6^A motifs that YTHDF1 bound to target mRNA. In addition, we conducted GO biological process analysis using DAVID to explore the biological functional role of YTHDF1-related downstream target genomes.

### 4.10. Statistical Analysis

All experiments were performed three times independently. To make comparisons between the two groups, we applied GraphPad Prism 8.0 software (GraphPad Software Inc., La Jolla, CA, USA). To determine the differences for multiple comparisons, a one-way analysis of variance (ANOVA) followed by Tukey’s test was carried out with treatments as independent factors. Student’s t-test was used for comparisons between the two groups. Data were calculated as mean ± standard deviation (SD). The hypothesis of nullity was rejected in all cases at the 0.05 level. A *p*-value <0.05 was considered statistically significant (* *p* < 0.05, ** *p* < 0.01, *** *p* < 0.001).

## 5. Conclusions

In conclusion, our study shows that hypoxia induces YTHDF1 expression in MC3T3-E1 cells and attenuates the inhibitory effect of hypoxia on osteogenesis by promoting its downstream THBS1 translation and stabilizing THBS1 mRNA. Taken together, the results indicate that m^6^A modification of mRNA plays a vital role in MC3T3-E1 cells under hypoxia, and YTHDF1 with its downstream THBS1 might provide potential candidates for the treatment of hypoxia-induced bone loss in peri-implantitis.

## Figures and Tables

**Figure 1 ijms-24-01741-f001:**
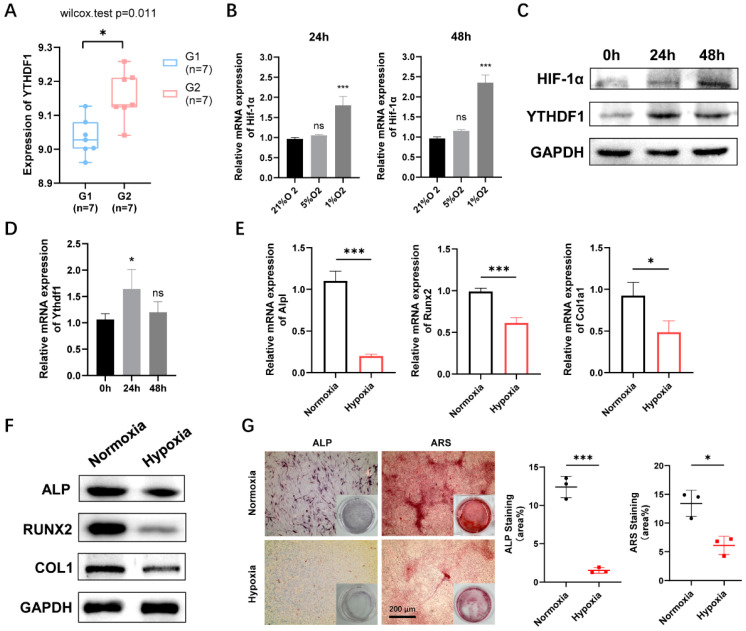
Hypoxia-induced osteogenesis downregulation and YTHDF1 promotion in MC3T3-E1 cells. (**A**) The gene expression profile dataset GSE33774 in GEO was selected to evaluate the differential expression of YTHDF1 in normal groups (**G1**) and peri-implantitis groups (**G2**). (**B**) The expression of Hif-1a at different O_2_ concentrations (21%, 5%, 1%) for 24 h and 48 h were determined by qRT-PCR. (**C**) The protein expression of HIF-1α and YTHDF1 at 1% O_2_ concentrations at different time points by Western blot. GAPDH was used as the internal control. (**D**) The mRNA expression of YTHDF1 under 1% O_2_ concentration conditions at different time points. (**E**) mRNA expression of osteogenic differentiation-related marker genes including *Alp*l, *Runx2*, and *Col1a1* under hypoxic treatment. (**F**) Protein level of ALP, RUNX2, and COL1 under hypoxia was verified by Western blot. (**G**) ALP staining and ARS staining showed reduced osteogenic ability in MC3T3-E1 cells under hypoxic conditions. Scale bar, 200 μm, n = 3. Results are presented as mean ± SD, Student’s *t*-tests, and one-way ANOVA followed by Tukey’s test (ns: not significant, * *p* < 0.05, *** *p* < 0.001, compared with the control group). YTHDF1, Yth m^6^A RNA-binding protein 1; HIF-1α, hypoxia-inducible factor-1α; ALP, alkaline phosphatase; RUNX2, runt-related transcription factor 2; COL1, type 1 collagen; ARS, alizarin red staining.

**Figure 2 ijms-24-01741-f002:**
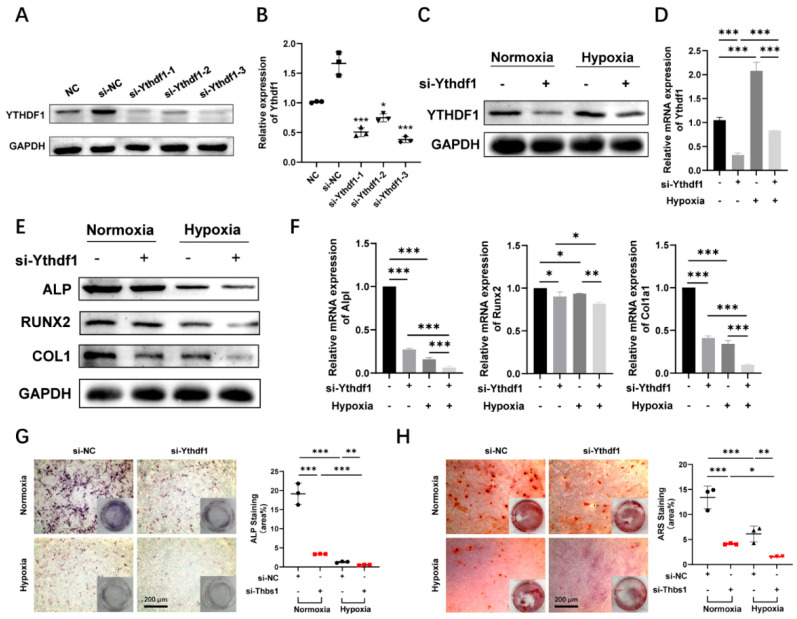
YTHDF1 promotes osteogenic differentiation under normoxic and hypoxic conditions. (**A**,**B**) YTHDF1 protein and mRNA expression determined through qRT-PCR and Western blot after 48 h transfection of siRNAs or negative control in MC3T3-E1 cells. (**C**,**D**) Protein and mRNA expressions of YTHDF1 in MC3T3-E1 cells transfected with silenced YTHDF1 were detected after 24 h cultured under hypoxia (1% O_2_). (**E**,**F**) mRNA and protein expression levels of ALP, RUNX2, and COL1 after transfected with silenced YTHDF1 were confirmed with qRT-PCR and Western blot in comparison with si-NC. (**G**,**H**) Examples of ALP versus ARS staining showed the inhibitory effect of knocking down YTHDF1 on differentiation of MC3T3-E1 osteoblasts cultured in the osteogenic medium under normoxic and hypoxic culture. Statistics show the reduced ALP activity of si-Ythdf1 relative to si-NC with positive staining for calcium nodules, as indicated by the elevated percentage of stained areas. Scale bar, 200 μm, n = 3. Results are presented as mean ± SD. The *p* values were calculated by one-way ANOVA followed by Tukey’s test (* *p* < 0.05, ** *p* < 0.01, *** *p* < 0.001, compared with the control group). YTHDF1, Yth m^6^A RNA-binding protein 1; ALP, alkaline phosphatase; RUNX2, runt-related transcription factor 2; COL1, type 1 collagen; ARS, alizarin red staining.

**Figure 3 ijms-24-01741-f003:**
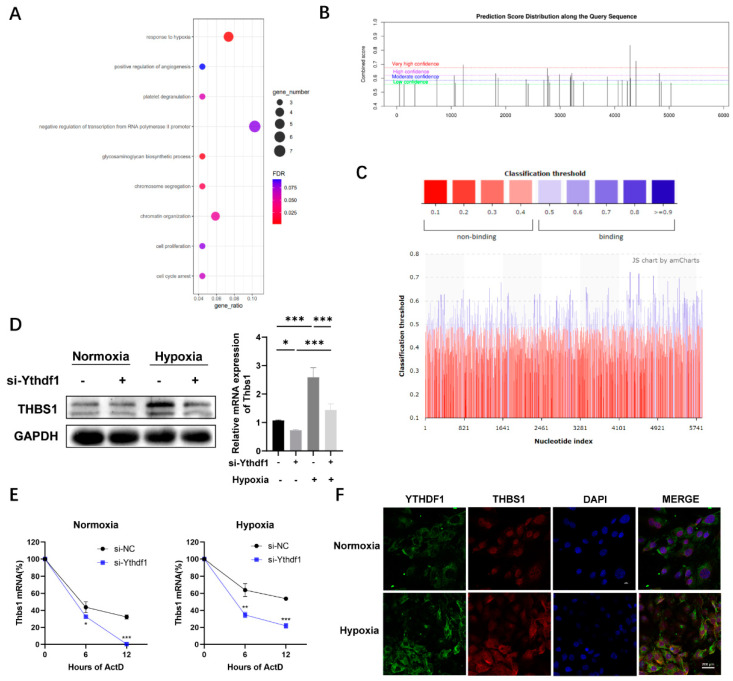
YTHDF1 mediates THBS1 mRNA stability via m^6^A modification. (**A**) Overlap of YTHDF1-binding peaks and m^6^A-containing peaks obtained from microarray data available at https://share.weiyun.com/xcB3waFn, go biological process enrichment analysis of the peaks overlap were performed using DAVID bioinformatics database. (**B**) SRAMP program predicted m^6^A sites in THBS1 mRNA. H: highly putative predicted m^6^A sites; V–H: very highly putative predicted m^6^A sites. (**C**) Prediction of binding sites between YTHDF1 protein and THBS1 mRNA by PRIdictor. (**D**) expression of THBS1 transfected with silenced YTHDF1 under normoxia and hypoxia was detected using Western blot and qRT-PCR. (**E**) RNA stability assays showed THBS1 mRNA of chondrocytes transfected with silenced YTHDF1. (**F**) The expression and location of YTHDF1 and THBS1 were measured by Immunofluorescence (IF) assays in MC3T3-E1 cells cultured for 24 h under normoxia (21% O_2_) or hypoxia (1% O_2_). Scale bar, 200 μm, n = 3. Results are presented as mean ± SD. The *p* values were calculated by one-way ANOVA followed by Tukey’s test (* *p* < 0.05, *** *p* < 0.001, compared with the control group). YTHDF1, Yth m^6^A RNA-binding protein 1; THBS1, thrombospondin-1; Act D, actinomycin D.

**Figure 4 ijms-24-01741-f004:**
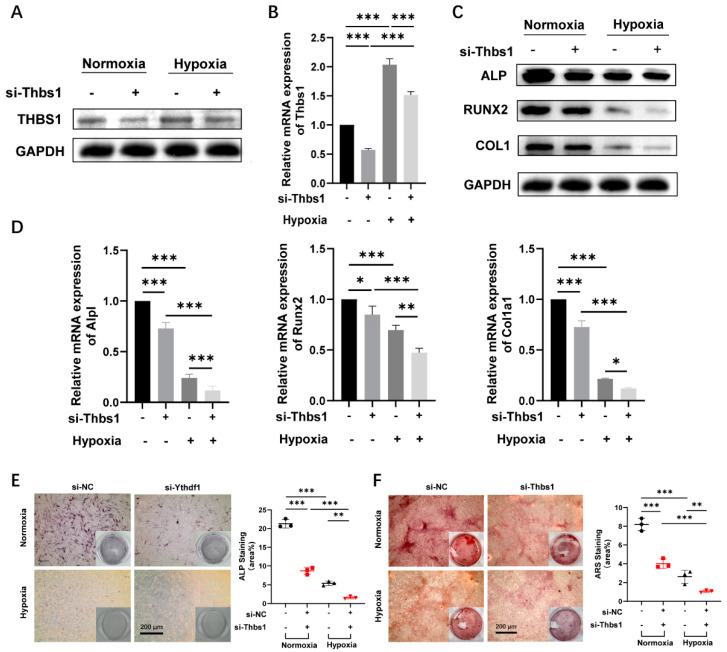
THBS1 promotes MC3T3-E1 osteogenic differentiation under hypoxic conditions. (**A**,**B**) Protein and mRNA expression of THBS1 in MC3T3-E1 cells transfected with silenced THBS1 were detected after 48 h cultured under hypoxia (1% O_2_). (**C**,**D**) Protein and mRNA expression levels of ALP, RUNX2, and COL1 after transfected with si-Thbs1 were detected using qRT-PCR and Western blot in comparison with si-NC under normoxia or hypoxia. (**E**,**F**) ALP and ARS staining show the inhibitory effect of si-Thbs1 knocking down THBS1 on differentiation of MC3T3-E1 osteoblasts cultured in the osteogenic medium under normoxic (21%) and hypoxic (1%) culture. The statistic shows the reduced ALP activity of si-Ythdf1 relative to si-NC with positive staining for calcium nodules, as indicated by the elevated percentage of stained areas. Scale bar, 200 μm, n = 3. Results are presented as mean ± SD. The *p* values were calculated by one-way ANOVA followed by Tukey’s test (* *p* < 0.05, ** *p* < 0.01, *** *p* < 0.001, compared with the control group). THBS1, thrombospondin-1; ALP, alkaline phosphatase; RUNX2, runt-related transcription factor 2; COL1, type 1 collagen; ARS, alizarin red staining.

**Figure 5 ijms-24-01741-f005:**
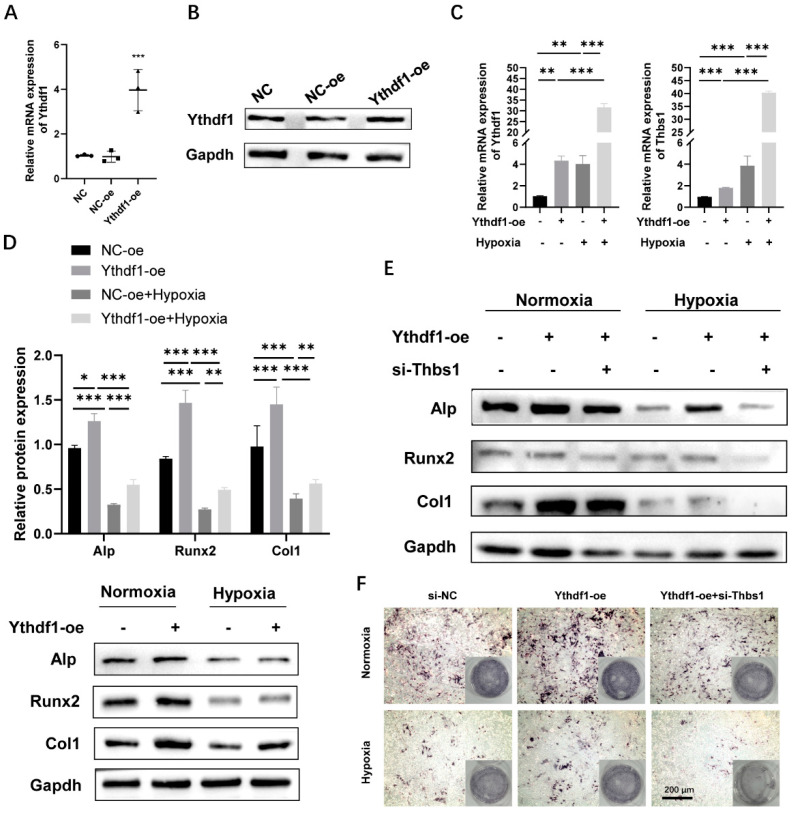
Overexpression of YTHDF1 promotes osteogenic differentiation of MC3T3-E1 cells via YTHDF1/THBS1 pathway under hypoxic conditions. (**A**,**B**) YTHDF1 mRNA and protein expression determined through qRT-PCR and Western blot after 48 h transfection of siRNAs or negative control in MC3T3-E1 cells. (**C**) mRNA expression of YTHDF1 and THBS1 were examined after overexpression of YTHDF1 under normoxia or hypoxia. (**D**) The protein expression levels of ALP, RUNX2, and COL1 after transfection with overexpressed YTHDF1 were detected by Western blot, compared with NC-oe under normoxia or hypoxia. (**E**) The protein expression levels of ALP, RUNX2, and COL1 after co-transfection of si-Thbs1 with Ythdf1-oe under normoxia or hypoxia. (**F**) ALP staining showed that overexpression of YTHDF1 promoted differentiation of MC3T3-E1 osteoblasts in both normoxic and hypoxic cultures, while the knockdown of THBS1 showed a slight regression of the osteogenic ability promoted by Ythdf1-oe. Scale bar, 200 μm, n = 3. Results are presented as mean ± SD. The *p* values were calculated by one-way ANOVA followed by Tukey’s test (* *p* < 0.05, ** *p* < 0.01, *** *p* < 0.001, compared with the control group). YTHDF1, Yth m^6^A RNA-binding protein 1; THBS1, thrombospondin-1; ALP, alkaline phosphatase; RUNX2, runt-related transcription factor 2; COL1, type 1 collagen.

**Figure 6 ijms-24-01741-f006:**
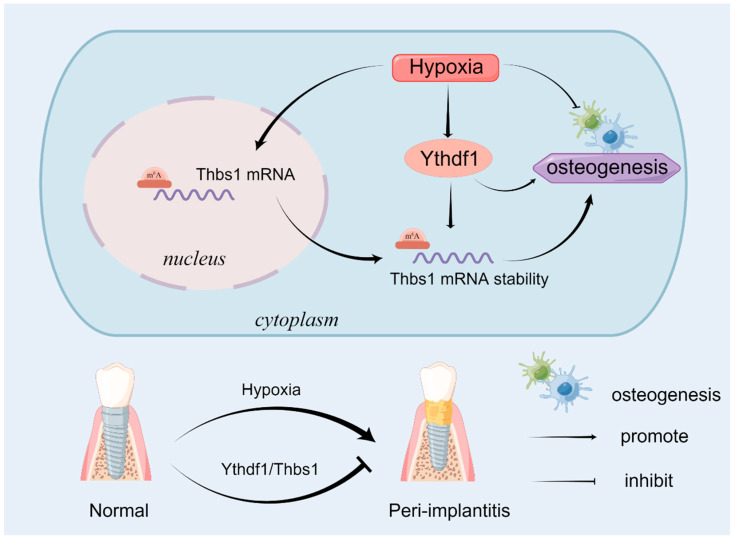
The role of the RNA-binding protein of N6-methyladenosine methylation YTHDF1 in a hypoxia-induced MC3T3-E1 cell model of osteoblasts. Hypoxia inhibited osteogenic differentiation and promoted the expression of YTHDF1 and THBS1, and the depletion of either YTHDF1 or THBS1 exacerbated the osteogenic inhibition caused by hypoxia, respectively. The increased expression of YTHDF1 under hypoxia enhanced the translation of THBS1 under hypoxia by stabilizing the mRNA transcript of THBS1. Inflammation-induced hypoxia aggravates bone resorption in peri-implantitis, while hypoxia-activated YTHDF1 with THBS1 promotes osteogenesis to prevent bone resorption. YTHDF1, Yth m^6^A RNA-binding protein 1; THBS1, thrombospondin-1.

**Table 1 ijms-24-01741-t001:** siRNA sequences for transcription.

siRNA	Sequences (5′→3′)
YTHDF1 siRNA no. 1	GCACTGACTGGTGTCCTTT
YTHDF1 siRNA no. 2	GGAAATGCCCAACCTACTT
YTHDF1 siRNA no. 3	GCACACAACCTCTATCTTT
THBS1 siRNA	S: UCAUCUGGUAUACCAUUGCTT AS: GCAAUGGUAUACCAGAUGAUA

**Table 2 ijms-24-01741-t002:** Primer sequences for qRT-PCR.

Gene	Forward Primer (5′→3′)	Reverse Primer (5′→3′)
*Hif-1α* (NM_001313919.1)	GAATGAAGTGCACCCTAACAAG	GAGGAATGGGTTCACAAATCAG
*Ythdf1* (NM_173761.3)	CCCTGTCCTGGAGAAACTGAAAGC	GTACTTGATGGAGCGGTGGATGTC
*Runx2* (NM_001145920.2)	TCACCTTGACCATAACAGTCTTCAC	TCTGTCTGTGCCTTCTTGGTTC
*Alpl* (NM_001287172.1)	GAACTGATGTGGAATACGAACTGG	TAGTGGGAATGCTTGTGTCTGG
*Col1a1* (NM_007742.4)	GCATGGCCAAGAAGACATCC	CCTCGGGTTTCCACGTCTC
*Thbs1* (NM_001313914.1)	GAAAGACGCCTGCCCAATTAAT	ACTTGATTTTCTGTCACATCGC
*β-actin* (NM_007393.5)	GCCAACCGTGAAAAGATGAC	ACCAGAGGCATACAGGGACAG

## Data Availability

Data sharing not applicable.

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
