# Peer review of "Yth m6A RNA-Binding Protein 1 Regulates Osteogenesis of MC3T3-E1 Cells under Hypoxia via Translational Control of Thrombospondin-1"

_ijms, 2023, doi:10.3390/ijms24021741_

Round 1

Reviewer 1 Report

MC3T3-E1 cells. The study is very complex and comprehensive. The graphical abstract at the end of the publication is positive. Otherwise, the manuscript is difficult to read because of the large number of abbreviations used. Here the authors should consider if there are ways to avoid this. The manuscript can be published after some changes.

Major Comments:

- A title without abbreviations should be chosen

- The working hypothesis should already be clearly stated in the abstract.

- In the figures, abbreviations should either be explained again in the legend, or written out in the graphs themselves.

- Normally, HIF1α is stabilized posttranscriptionally. Why are differences in mRNA seen here. The authors should comment on this.

- The graphs should be arranged according to the ABC, this improves the clarity of the figures

- The statistics used should be stated in the figure legends.

- Why was actin chosen as the reference gene for qPCR? Has this been validated previously?

- More information should be given about the primer design.

- The statistics are unclear. Since the number of cases is very small with n=3 a non-parametric assay should be used. Why and when was paired/unpaired compared? This must be stated and graphically evident.

Minor comments

- In the legend to Fig. 1C, HIF1α is missing.

- Some abbreviations are not introduced.

- Which E1 subtype was used?

- The order numbers for the antibodies are missing.

- murine genes should be written in italics and proteins in capital letters.

Reviewer 2 Report

This study identifies a link between YTHDF1 and ostoegenic differentiation in normoxia and hypoxia. For the most part all the data is clear and experiments are properly controlled. Just have a few things that need clarifying.

1. the authors indicate that YTHDF1 is induced in hypoxia both at RNA and protein level, however, knockdown of this protein does not reverse the hypoxia phenotype. This needs at least to be discussed.

2. Similar to this is the case for Thbs1.

3. Thbs1 RNA stability is altered by YTHDF1, is this also the case in hypoxia? The data should be there, but was not presented.

Round 2

Reviewer 1 Report

The authors implemented my suggestions to my satisfaction.

Reviewer 2 Report

The authors addressed my main concerns